# Canagliflozin Inhibited the Activity of Hemolysin and Reduced the Inflammatory Response Caused by *Streptococcus suis*

**DOI:** 10.3390/ijms241713074

**Published:** 2023-08-22

**Authors:** Xiaodan Li, Qingyuan Li, Zhaoran Zhang, Chenchen Wang, Xinyu Huo, Hongjiang Lai, Hao Lu, Wenjia Lu, Yulin Qian, Wenqi Dong, Chen Tan, Manli Liu

**Affiliations:** 1Hubei Biopesticide Engineering Research Centre, Wuhan 430000, China; xiaodanli_0721@163.com; 2College of Veterinary Medicine, Huazhong Agricultural University, Wuhan 430000, China; lqy1111@webmail.hzau.edu.cn (Q.L.); zhangzhaoran@mail.hzau.edu.cn (Z.Z.); 2018302110164@webmail.hzau.edu.cn (C.W.); 2020302110145@webmail.hzau.edu.cn (X.H.); laihongjiang@webmail.hzau.edu.cn (H.L.); 88251420@webmail.hzau.edu.cn (H.L.); 2017302110131@webmail.hzau.edu.cn (W.L.); qianyulin@webmail.hzau.edu.cn (Y.Q.); dongwq@mail.hzau.edu.cn (W.D.); tanchen@mail.hzau.edu.cn (C.T.)

**Keywords:** *Streptococcus suis*, hemolysin, canagliflozin, anti-hemolysin

## Abstract

Highly virulent *Streptococcus suis* (*S. suis*) infections can cause Streptococcal toxic shock-like syndrome (STSLS) in pigs and humans, in which an excessive inflammatory response causes severe damage. Hemolysin (SLY) is a major virulence factor of *S. suis* serotype 2 that produces pores in the target cell membrane, leading to cytoplasmic K^+^ efflux and activation of the NLRP3 inflammasome, ultimately causing STSLS. The critical aspect of hemolysin in the pathogenesis of *S. suis* type 2 makes it an attractive target for the development of innovative anti-virulence drugs. Here, we use the *S. suis* toxin protein (SLY) as a target for virtual screening. A compound called canagliflozin, a hypoglycemic agent, was identified through screening. Canagliflozin significantly inhibits the hemolytic activity of hemolysin. The results combined with molecular dynamics simulation, surface plasmon resonance, and nano differential scanning fluorimetry show that canagliflozin inhibits the hemolytic activity of SLY by binding to SLY. In addition, canagliflozin markedly reduced the release of SC19-induced inflammatory factors at the cellular level and in mice. Importantly, the combination of canagliflozin and ampicillin had a 90% success rate in mice, significantly greater than the therapeutic effect of ampicillin. The findings suggest that canagliflozin may be a promising new drug candidate for *S. suis* infections.

## 1. Introduction

The use of antibiotics is one of the important means to reduce human and animal mortality [1]. Bacterial resistance is an inevitable phenomenon. However, the development of new antibiotics faces many problems, including a difficult and long development cycle and the difficulty in getting a return on investment in the short term [2]. Worryingly, the pace of new drug development is far slower than the rate at which bacteria develop resistance. China is a large livestock country and the use of antibiotics is extensive, which has led to an increase in animal-derived drug-resistant bacteria. Thus, the development of new antimicrobial agents is a more important issue there.

*Streptococcus suis* (*S. suis*) is an important zoonotic pathogen and is still the most common pathogen found in swine farms [3,4]. Pigs or humans infected with *S. suis* experience sepsis, meningitis, and even acute inflammatory storms [5,6]. Of the 29 *S. suis* serotypes, *S. suis* 2 (SS2) is the most prevalent and virulent in pigs and humans. The SS2 outbreak in China in 2005 resulted in more than 200 cases of human infection, with a mortality rate of 20% [7]. The *S. suis* SC19 strain was isolated from the pigs during the 2005 *S. suis* outbreak in Sichuan Province and maintained in the laboratory [8]. Infection with the prevalent *S. suis* strain SC19 induces an acute and extremely high inflammatory cytokine response, with SC19 causing much higher levels of inflammation and organ damage than the classical virulent strain P1/7, which can also lead to high mortality [5,6,9]. In previous studies, ampicillin had a good therapeutic effect for *S. suis* infection after 1 h treatment, but the therapeutic effect was greatly reduced after 6 h of treatment with ampicillin [10]. Inflammatory storms lead to the failure of antibiotic treatment. Hemolysin protein (SLY) plays a major role in the generation of the *S. suis* inflammatory storm [5]. In 1994, the SLY of *S. suis* was first defined and identified as a key pathogenic virulence factor [11,12]. SLY is a member of the cholesterol-dependent cell hemolysin (CDC) family [13]. After being released by bacteria, hemolysin binds to the surface of host cells to form holes and cause STSLS. The discovery of some drugs [14,15,16,17] with anti-SLY activity gave us a good indication that treatment at 6 h after infection would significantly improve the survival rate of mice. The screening of drugs that target bacterial virulence proteins can have great advantages [18,19].

Drugs from the FDA drug library have the advantage of high safety and greatly reduced development time compared to other drugs [20]. Currently, an increasing number of previously discovered drugs are being found to have significant antibacterial effects. The antirheumatic drug auranofin has been shown to have a significant bactericidal effect on Gram-positive bacteria [21] and, in combination with colistin and meropenem, to treat sepsis infections caused by multidrug-resistant Gram-negative bacteria [22]. The antiparasitic drug oxyclozanide has a significant bactericidal effect on Gram-positive bacteria [23]. Metformin, a classic antidiabetic drug with antibacterial and anti-biofilm activities [24], has promising application prospects as a tetracycline adjuvant in vitro and in vivo, providing a new treatment plan for the treatment of tetracycline-resistant bacteria [25].

Inspired by these results, we used a virtual screening approach to screen for drugs that target SLY proteins in the FDA-approved drug library. From the screening results, we further identified a hypoglycemic drug called canagliflozin. It is worth noting that it has good antihemolytic activity and reduces the inflammatory response triggered by *S. suis*. Combined treatment with canagliflozin and ampicillin, had a good therapeutic effect. Here, we once again demonstrate the reliability of drugs targeting toxin proteins in the treatment of *S. suis*. Virtual screening improves the effectiveness of our screening program and has great potential for use in drug research and development.

## 2. Results

### 2.1. Screening for Compounds Targeting Hemolysin

We performed a virtual screening of the FDA-marketed drug library by molecular docking. A total of 1004 compounds were screened, and 15 compounds were screened using the calculated receptor-ligand binding affinity as the ordering index (Figure 1A). The virtually screened drugs were further screened for anti-hemolysis activity in vitro. A new round of screening was performed to identify compounds with >50% inhibition of SLY protein hemolysis at a concentration of 64 μg/mL (Appendix A). Canagliflozin was identified as the most promising antihemolytic drug (Figure 1B).

### 2.2. Canagliflozin Inhibits the Activity of the S. suis Hemolysin Protein SLY

We identified canagliflozin by targeting the SLY protein of *S. suis* as a screening compound. The supernatant of *S. suis* SC19 contains SLY, which can have a hemolytic effect on red blood cells. We evaluated the antihemolytic effect of canagliflozin using the supernatant of SC19 cultures. The inhibitory effect of canagliflozin on the hemolysis capacity of the SC19 supernatant became more evident with increasing canagliflozin concentration. When the concentration of canagliflozin was higher than 64 μg/mL, the inhibition of hemolysis reached more than 50% (Figure 1C). At the same time, the SLY protein was recombinantly expressed and purified, and 912.5 ng/mL of SLY produced 97.8% hemolysis in 1% erythrocytes. Canagliflozin inhibited the hemolytic activity of SLY by 76% (Figure 1D) at a concentration of 16 μg/mL.

### 2.3. Canagliflozin Activity against S. suis In Vitro

We tested the antibacterial activity of canagliflozin against *S. suis* by MIC experiments (Table 1). Canagliflozin showed weak antibacterial activity and inhibited the growth of *S. suis* at a concentration of 128 μg/mL. We evaluated the MIC of standard strains of *Staphylococcus aureus* (*S. aureus*), *Listeria monocytogenes*, *Bacillus subtilis*, and *Streptococcus pneumoniae*, and the results showed that canagliflozin had a bacteriostatic effect *against* bacteria at 64–128 μg/mL. Canagliflozin also has similar antibacterial activity in clinical strains of *S. aureus* and *S. suis*. These results demonstrated that a high concentration of (128 μg/mL) canagliflozin had a bacteriostatic effect on Gram-positive bacteria. We further examined the growth curves of *S. suis* at 16 μg/mL, 32 μg/mL, and 64 μg/mL canagliflozin. Figure 2A shows the activity of canagliflozin at 1/2 and 1/4 MIC concentrations, which slowed *S. suis* bacterial growth.

In addition, we conducted a preliminary safety evaluation of canagliflozin in Vero cells, RAW264.7 cells, and sheep red blood cells. Canagliflozin did not show obvious cytotoxicity to Vero cells at 64 μg/mL (Figure 2B), and similar results were found in RAW264.7 macrophages (Appendix A). The results of the hemolytic toxicity of canagliflozin showed that there was no hemolytic toxicity of canagliflozin at 128 μg/mL (Figure 2C). These results suggest that canagliflozin has no additional toxicity at the concentrations required for antimicrobial activity and inhibition of the hemolytic activity of SLY.

### 2.4. Canagliflozin Reduced the Inflammatory Response of RAW264.7 Cells Induced by SC19

When *S. suis* with high SLY expression attacks host cells, a large number of inflammatory factors are produced due to increased membrane perforation, resulting in a large number of acute inflammatory injuries. We infected RAW264.7 mouse macrophages with SC19. By measuring transcript and protein levels, an inflammatory model was obtained, and the secretion of IL-1β and TNF-α by the cells was increased. After SC19 infection, we added canagliflozin for 6 h of treatment. Figure 3 shows that the secretion levels of IL-1β and TNF-α were significantly reduced in the canagliflozin treatment group. At the transcription level, the mRNA levels of IL-1β and TNF-α decreased with increasing concentrations of canagliflozin (Figure 3A,B). Similar results were obtained at the protein level (Figure 3C,D). Canagliflozin had a good anti-inflammatory effect on RAW264.7 cells infected with SC19 at 32 μg/mL. Studies have shown that IL-1β secretion is a factor that further causes inflammatory cytokine storms, and IL-1β secretion was dependent on the decrease in the net concentration of canagliflozin after infection. Therefore, we conclude that canagliflozin not only reduces the activity of SLY but also reduces the release of cellular inflammatory cytokines, which further reduces the inflammatory storm in the host caused by strains that highly express SLY.

### 2.5. Determination of the Canagliflozin Targeting of the S. suis Hemolysin Protein In Vitro

To further investigate the inhibitory activity of canagliflozin on SLY, we used molecular dynamics simulations to explore their interactions. The root mean square deviation (RMSD) of the complex fluctuated between 0.1 nm after 4 ns, and the last 1 ns of the simulation was used for subsequent analysis (Figure 4A). The results of the molecular simulation show that electrostatic and van der Waals forces play a key role in the interaction of canagliflozin and hemolysin proteins. As shown in Figure 4B, the canagliflozin molecule is located in SLY domain 2. Canagliflozin binds to ASN50, GLU53, TYR54, and SER374 in domain 2 by electrostatic and van der Waals interactions (Figure 4C), which most likely prevents the late conformational change in domain 2 and leads to the loss of SLY function.

In addition, the in vitro binding capacity of the SLY protein to canagliflozin was determined. The temperature stability of the SLY protein was investigated by Nano DSF. Protein stability changes with the addition of SLY to canagliflozin, with denaturation initiation temperatures decreasing by 6.10 °C and tm values decreasing by 3.80 °C (Figure 4D). The results showed that the binding of canagliflozin to the SLY protein resulted in a higher susceptibility of the SLY protein to degeneration. The figure shows that the binding of canagliflozin to SLY proteins causes SLY proteins to more readily change from a folded to an unfolded state at elevated temperatures, resulting in greater exposure of the embedded tryptophan. At the same time, the SPR assay demonstrated that canagliflozin has a strong binding with SLY. As shown in Figure 4E, the equilibrium dissociation constant (KD) of canagliflozin and SLY was determined to be 0.90 µM. By molecular dynamics simulations and in vitro binding analyses, we demonstrated that canagliflozin specifically binds to the SLY protein.

### 2.6. Canagliflozin Improved the Therapeutic Effect of Ampicillin on SC19 Infection In Vivo

β-lactam antibiotics have a good early treatment effect in *S. suis* with a high SLY expression. In the late stage of bacterial infection, antibiotic treatment can effectively kill the bacteria, but severe inflammatory damage caused by SLY and other virulence factors also leads to host death. We investigated the efficacy of canagliflozin in the treatment of mice infected with SC19 for 6 h. A mouse peritonato-septicemia model was used to evaluate the therapeutic effect of canagliflozin (Figure 5A). To induce infection, 2.5 × 10^8^ SC19 was injected intraperitoneally in each mouse. Six hours after infection, animals were treated with an intraperitoneal injection of canagliflozin (5 mg/kg), ampicillin (10 mg/kg), the combination (5 + 10 mg/kg), and PBS (bacterial bearing groups and control group). Acute death occurred at 48 h in the untreated and ampicillin groups. The canagliflozin treatment provided short-term protection to the mice within 48 h because canagliflozin did not have good antibacterial activity, and the mice still died after 2 days of treatment. The combined treatment with canagliflozin and ampicillin greatly improved the survival rate of mice in the later stage of infection. The body weight of the mice was monitored throughout the experiment and the results showed severe weight loss in SC19 infection mice. It was observed that the weight of canagliflozin-treated mice decreased by 16% within 3 days of infection and increased from day 4 (Figure 5B). At the same time, we evaluated the clinical symptoms of the surviving mice, and the results showed that the clinical signs of the mice treated with the combination group were significantly improved (Figure 5C).

The therapeutic effect of canagliflozin on the host was evaluated by infecting mice with a sublethal dose of 1 × 10^8^ bacteria. The amount of bacteria in organs, pathological injury, and blood inflammation were measured 6 h after drug treatment (Figure 5D–G). Compared with the untreated group, the ampicillin group and the combination group had significantly reduced amounts of bacteria in the organs, which were reduced by 2 lg. There was no significant change in the combination group compared with the ampicillin group. The results showed that the addition of canagliflozin did not increase the ampicillin activity against bacteria. Treatment with canagliflozin alone had no effect on bacterial load. The results showed that the levels of IL-1β and TNF-α were significantly decreased in the canagliflozin alone and combination groups (Figure 6A,B). These experiments demonstrated that canagliflozin has a significant anti-inflammatory effect in mice infected with *S. suis* with a high SLY expression.

Compared with the untreated group, lung congestion, inflammatory cell infiltration, severe spleen congestion, and liver vacuolar degeneration and necrosis were slightly reduced in the ampicillin-treated group (Figure 6C). Surprisingly, canagliflozin alone and in combination significantly reduced the severity of pathological changes in the lungs, liver, and spleen of mice. Compared with ampicillin alone, the addition of canagliflozin significantly reduced pathological damage and the levels of inflammatory factors, which also explains why the success rate of ampicillin and canagliflozin combination treatment was greatly improved.

The addition of canagliflozin effectively prevents deaths due to inflammatory storms when antibiotics are used to treat late stages of infection caused by *S. suis* with a high SLY expression.

## 3. Discussion

As an important zoonotic pathogen, *S. suis* poses a serious threat to food safety and human public health and causes huge economic losses to the livestock industry [26,27]. Furthermore, the pace of development of new antibiotics is much slower than the pace of development of bacterial resistance, and new anti-infection strategies and mechanisms of drug discovery are urgently needed [18]. In contrast to the mechanism of action of conventional antibiotics, virulence factors play an important role in bacterial pathogenesis, and the anti-virulence strategy selects for the role of virulence factors that do not exert survival pressure on bacteria, thus slowing the development of bacterial resistance to some extent [28,29]. It has been shown that SLY, a virulence factor produced by *S. suis*, plays an important role in the infection process and that knockdown of SLY significantly reduces bacterial virulence [30], so selecting for inhibition of SLY may be essential in the treatment of *S. suis* infections.

In this study, the FDA-approved drug canagliflozin was found to bind well to the protein in a virtual screen using SLY as a target. In vitro experiments further demonstrated that canagliflozin effectively inhibited the hemolytic activity of bacterial SLY. High concentrations of canagliflozin had antibacterial effects on bacteria, and the concentrations were greater than the effective concentration needed to inhibit hemolysin activity. Thus, canagliflozin has an advantage over conventional antibiotics in terms of bacterial survival pressure. The molecular mechanism was investigated, and it was shown that canagliflozin did not affect the expression of the SLY protein but it bound the protein directly through electrostatic and van der Waals forces bonding, which was the predominant factor responsible for the anti-hemolysin activity of canagliflozin. In addition, as an FDA-approved hypoglycemic agent, certain in-depth safety and pharmacokinetic information is known for canagliflozin, which greatly reduces the cost and time needed to develop this drug and accelerates its translation into clinical treatment [31,32]. Therefore, canagliflozin is a rather promising candidate for the treatment of *S. suis*-associated infections.

Inflammation is part of the body’s first line of defense against aggressive pathogens. An appropriate inflammatory response ensures proper resolution of inflammation and removal of harmful stimuli, but excessive inflammation can lead to damage to surrounding normal cells and even the development of fatal disease [33,34]. The severity of *S. suis* infection is related to the host’s natural immune response, and the host’s immune system can release a variety of proinflammatory cytokines upon *S. suis* stimulation, which in turn triggers an excessive inflammatory response that accelerates the disease process [35,36,37]. The SLY protein plays an important role in the induction of excessive inflammatory response in the host, especially the excessive inflammatory response leading to STSLS [5]. Thus, alleviation of the excessive inflammatory response is critical in the management of late-stage *S. suis* infections. In this study, we investigated the effect of canagliflozin on post-inflammatory factors associated with *S. suis* infection and found that canagliflozin dose-dependently and significantly inhibited the accumulation of inflammatory factors in RAW264.7 cells induced by *S. suis* SC19. In addition, treatment with canagliflozin in combination with the clinically used antibiotic ampicillin reduced the level of inflammatory factors, bacterial load, and histopathologic damage in *S. suis* SC19-infected mice.

Although conventional antibiotics are effective in eradicating *S. suis*, these antibiotics do not improve the survival of infected mice by suppressing excessive proinflammatory responses. Here, we found that ampicillin had a low rate of protection (10%) against nonresistant *S. suis* in a model of severely infected mice. However, the addition of canagliflozin resulted in a significantly higher survival rate (90%). The addition of canagliflozin reduced the inflammatory response and ultimately significantly improved clinical signs in mice compared to ampicillin alone. The current results suggest that the neutralization of SLY toxicity and the reduction in the inflammatory response by canagliflozin are the main reasons for this phenomenon. Therefore, in some patients with severe infections, canagliflozin in combination with β-lactams may be a better choice. Excitingly, the combination of canagliflozin and ampicillin did not increase hematologic activity and cytotoxicity, which is a key issue in clinical combination therapy. In conclusion, these findings suggest that an anti-SLY treatment strategy is a potential novel approach to controlling *S. suis* infections. Canagliflozin is used in the treatment of type 2 diabetes in humans at doses of 100–300 mg/day [38]. No significant cumulative toxicity has been observed with canagliflozin over a long period of time [39,40]. In our study, a lower therapeutic dose than that used in humans was used in the mouse experiment and administered only once after infection. These results indicate that canagliflozin has a high potential value in the treatment of *S. suis* infections. Meanwhile, these data also lay the groundwork for the development of canagliflozin as a potential novel drug for the treatment of *S. suis* infections.

## 4. Materials and Methods

### 4.1. Bacterial Strains, Growth Conditions, and Preparation of Chemicals

The *S. suis* SC19 strain was isolated from the brains of pigs during the 2005 *S. suis* outbreak in Sichuan Province and maintained in the laboratory. Other strains of *S. suis* and *S. aureus* were isolated and preserved in the laboratory. *S. suis* was grown in tryptic soy broth (TSB) or inoculated onto tryptic soy agar (TSA) (Summus Ltd., Shanghai, China) supplemented with 10% newborn bovine serum (NBS) (Sijiqing Ltd., Shanghai, China) at 37 °C. Mouse RAW264.7 macrophage-like cells were grown in Dulbecco’s modified Eagle’s medium (DMEM) (Gibco, Gibco, Grand Island, NY, USA) with additional high glucose, and 10% heat-inactivated fetal bovine serum (FBS) (Gibco, Waltham, MA, USA). Drugs were obtained from TargetMol (Shanghai, China).

### 4.2. Screening Method

Proteins were obtained from the RCSB database, 3HVN structures were selected, and solvents and ligands were removed using MOE (version 2009). The small-molecule structure was obtained from the PubChem database, and Pipeline Pilot (version 8.5) was used to construct the small-molecule pretreatment process, including small-molecule hydrogenation, chiral center determination, and rotatable bond determination. The virtual screening was performed by smina (version 2020), which is a software based on AutoDock Vina (version 1.5.7). The docking domain covered the entire structure of the receptor. The screening generated 20 different conformations for each small molecule and ligand. Each conformation was docked separately, and the 5 conformations with the highest docking score were selected as outputs. The scoring function used default parameters.

### 4.3. Bacterial Culture

*S. suis* SC19 preserved bacterial solution was inoculated on a TSA solid plate containing 10% NBS. The selected single colonies were transferred to a liquid TSB medium containing 10% NBS and allowed to grow. The activated bacterial suspension was transferred to a new medium at a ratio of 1:100 to grow to OD600 nm = 0.6 to be used later in experiments.

### 4.4. Minimum Inhibitory Concentration Test

The MIC experiment was performed using standard broth microdilution according to CLSI guidelines. Canagliflozin was diluted 2 times with MH medium containing 5% FBS and mixed with the same volume of bacterial working fluid. The final bacterial concentration was 5 × 10^5^ CFU/mL. The mixed system was added to a 96-well plate and incubated at 37 °C for 18 h. The criterion for MIC is the absence of visible turbidity. There were three replicate wells for each sample.

### 4.5. Measurement of Growth Curves

*S. suis* SC19 grown to OD600 nm = 0.6 was diluted to 1 × 10^6^ CFU using MH medium containing 5% NBS. Canagliflozin was diluted in the same medium and mixed with the diluted bacteria in an equal volume. Then, 200 μL of the mixture was added to a 100-well microplate. Bioscreen (FLUOstar Omega, Offenburg, Germany) was used to measure the OD600 nm values of bacteria at different time points. There were three replicate wells for each sample.

### 4.6. Recombinant SLY Expression and Purification

Recombinant SLY was expressed using the pET28a (+)-based *Escherichia coli* BL21 (DE3) expression system and purified by Ni-NTA. First, SLY sequences were subcloned into pET-28a (+) vectors using BamHI and SalI cleavage sites to construct the plasmid pET-28a (+)-SLY. Then, the recombinant plasmid was electroporated to *E. coli* BL21 (DE3), and the positive colonies were selected, inoculated into LB liquid medium containing kanamycin, and incubated at 200 rpm and 37 °C overnight. The bacterial suspension overnight culture was transferred to 1 L LB liquid medium containing kanamycin at a ratio of 1:100 and cultured (200 rpm, 37 °C) for approximately 3 h to the logarithmic growth stage (OD600 nm = 0.6). After adding 0.5 mM isopropyl-β-d-thiogalactoside (IPTG) to the medium, the culture was incubated continuously at 16 °C for 16 h to induce protein expression. Bacteria were harvested by centrifugation at 10,000 rpm for 5 min at 4 °C, resuspended in 50 mL PBS, and disrupted with a high-pressure cell crusher. After crushing, the mixture was centrifuged at 4 °C at 12,500 rpm for 30 min. The supernatant was filtered through a 0.45 μm filter (Millipore, Billerica, MA, USA) and added to a Ni-NTA column. PBS solution containing imidazole with different concentration gradients (0 mM, 10 mM, 30 mM, 50 mM, 100 mM, 200 mM, 500 mM) was applied for the linear gradient elution of proteins. Protein samples were collected, and the SLY protein was dialyzed three times at 4 °C with 2 L of 10 mM PBS (pH = 7.4). The concentration of SLY was assessed by using the BCA protein assay kit (Beyotime, Shanghai, China).

### 4.7. Determination of Antihemolytic Activity

Canagliflozin’s antihemolytic activity was measured as described previously [15]. *S. suis* SC19 from overnight culture was centrifuged to obtain the supernatant. The supernatant was filtered through a 0.22 filter (Millipore, Billerica, MA, USA). Various amounts of supernatant were added to defibrillated sheep erythrocytes. The minimum concentration that caused complete lysis of the red cells was selected. Using the same method, the hemolytic activity of the purified SLY protein was determined, and a concentration of 0.912 μg/mL was set as the dose for subsequent experiments. Canagliflozin solutions of different concentrations (0, 2, 4, 8, 16, and 32 μg/mL) were mixed with supernatant or 0.912 μg/mL SLY at 37 °C for 30 min. Then, a solution of 2% sheep red blood cells was added and incubated with the previous mixture at 37 °C for 30 min. Then we centrifuged the incubated cultures and collected the supernatant. The amount of heme in the supernatant was quantified by measuring the absorbance at 543 nm using a microplate reader. There was 100% heme release in red blood cells treated with 2.5% Triton X-100. The percentage of heme released in the sample is used to evaluate the anti-hemolysis effect of the drug.

### 4.8. Cell Infection Study

RAW264.7 cells were inoculated into 12-well cell culture plates at 5 × 10^5^ cells per well. A culture medium without antibiotics was used to inoculate the cells. SC19 diluted with DMEM was added to the cell pore plate (MOI = 10 or 25) when cell growth converged to 80%. Bacteria were gently washed with PBS after infection with SC19 for 1 h. Cells were then incubated for 6 h with DMEM containing 8, 16, and 32 μg/mL canagliflozin. The supernatant was used to determine the secretion of IL-1β and TNF-α by ELISA (Absin, Shanghai, China).

### 4.9. Toxicity Test

The Cell Counting Kit-8 (Beyotime, Shanghai, China) was used to determine cytotoxicity. Vero cells and RAW264.7 cells were added separately to a 96-well plate and each well contained 1 × 10^4^ cells. When the growth of the cells converged to 80%, 100 µL of different concentrations of canagliflozin or ampicillin were added. Absorbance was measured with an enzyme-labeled method on an instrument for the 2 h reaction.

Hemolytic toxicity was determined by measuring the percentage of heme released after the lysis of sheep red blood cells. Sheep red cells were sterile and had been washed to remove their fibrin with PBS. Different concentrations of canagliflozin or ampicillin were mixed with 2% sheep red blood cells in equal volumes and the samples were incubated for 30 min. The supernatant after centrifugation was collected and added to a 96-well plate. The absorbance was measured at a wavelength of 543 nm.

### 4.10. Nano Differential Scanning Fluorimetry

The Nano DSF experiment was conducted using Prometheasn.48 [41] (NanoTemper Technologies, München, Germany). SLY (197 µg/mL) was incubated with 16 µg/mL canagliflozin in PBS buffer for 20 min. The samples were tested using standard capillary tubes. The samples were exposed to conditions ranging from 20 °C to 100 °C at a thermal rise rate of 4 °C/min. Fluorescence from tryptophan after UV excitation at 280 nm was collected by detectors at 330 nm and 350 nm. Thermal stability parameters, including T_onset_ and T_m_, were calculated using PR.ThermControl software (version 2.1.5).

### 4.11. Molecular Dynamics Simulation

The kinetic simulation was performed with gromacs (version 2018.8), and the system was wrapped in a flexible SPC water model. The force field of the complex system was the AMBER force field, and the charge of the complex was balanced by Na+ ions. Temperature and pressure equilibria were then performed at 310 K (physiological ambient temperature) and 1 bar, respectively. On this basis, the final simulation was performed with a total of 6 ns and a step size of 2 fs, using SHAKE and particle mesh ewald (PME) to handle the motion and long-range electrostatic effects of all hydrogen-containing chemical bonds, respectively. GMX_MMPBSA (version 1.5.7) was used to analyze the binding free energy during the final simulation.

### 4.12. Surface Plasmon Resonance (SPR)

The protein binding capacity of SLY to canagliflozin was determined using BiacoreT200 [42] (Cytiva, Eysins, Switzerland). SLY protein was diluted to a final concentration of 50 μg/mL using sodium acetate solution (pH 3.0). The SLY protein is attached to the CM5 sensor chip by amino coupling (Cytiva, Eysins, Switzerland). Different concentrations of canagliflozin were injected at a flow rate of 30 μL/min, and the system was run in PBS-T (10 mM phosphate buffer with 0.05% Tween 20 and 5% DMSO). Data analysis was carried out using Biacore T200 v.3.0 software. The interaction between sly and canagliflozin was fitted using a steady-state affinity model.

### 4.13. Animal Studies

The experimental procedure for the infectivity of SC19 in mice was modified in accordance with a previous method. Female BALB/c mice aged 5–6 weeks (20–22 g) were randomly divided into 10 mice per group. After centrifugation, 2.5 × 10^8^ CFU was injected intraperitoneally, and 200 μL PBS was injected in the control group. Six hours after SC19 infection, mice were injected intraperitoneally with 200 µL of canagliflozin (5 mg/kg), ampicillin (10 mg/kg), or the combination (5 + 10 mg/kg). Clinical symptoms and survival of the infected mice were monitored twice daily for the first 2 days and once daily for the next 5 days. The weight change of the mice over 7 days was measured at 24 h intervals. Clinical symptoms were assigned clinical scores according to the following criteria [5]: 0 = normal response to stimuli; 1 = wrinkled coat, slow response to stimulation; 2 = to respond only to repeated stimuli; 3 = No reaction or walking in circles; 4 = death.

To evaluate the therapeutic effect of canagliflozin, mice were infected with a sublethal dose of SC19. Each mouse was injected intraperitoneally with 1 × 10^8^ CFU of SC19, and there were five mice in each group. Six hours after infection, mice were injected intraperitoneally with 200 μL of canagliflozin (5 mg/kg), ampicillin (10 mg/kg), or a combination (5 + 10 mg/kg). Orbital blood sampling and aseptic collection of the liver, lung, and spleen were performed on anesthetized mice 12 h after infection. After a series of 10-fold dilutions in PBS, the weighed and homogenized organs were inoculated with 5% NBS MHA. Colony counts were performed to calculate the amount of bacteria carried by the organs after incubation at 37 °C for 18 h. Serum was collected and centrifuged to determine IL-1β and TNF-α levels. For pathologic examination, the liver, lung, and spleen were fixed in 4% paraformaldehyde. Tissues were stained with hematoxylin and eosin after paraffin embedding.

### 4.14. Statistical Analysis

The mean ± standard deviation (SD) was used to express all experimental data. The two-tailed unpaired *t* test method and two-way ANOVA method were utilized for the analysis of experimental data by using GraphPad Prism 8.0 software.

## 5. Conclusions

In our study, the virtual screening of target drugs was used as an effective screening method. Canagliflozin not only targets the SLY of *S. suis* but also reduces the inflammatory response caused by *S. suis*. In vitro experiments, canagliflozin inhibited hemolysin activity by up to 100% and dose-dependently attenuated SC19-induced inflammatory cytokine release at the cellular level. In animal studies, we have shown that canagliflozin can reduce the inflammatory response induced by SC19 and significantly enhance the therapeutic effect of ampicillin six hours after infection. Our study highlights the important role of antiviral drugs in the treatment of *S. suis* infections.

## Figures and Tables

**Figure 1 ijms-24-13074-f001:**
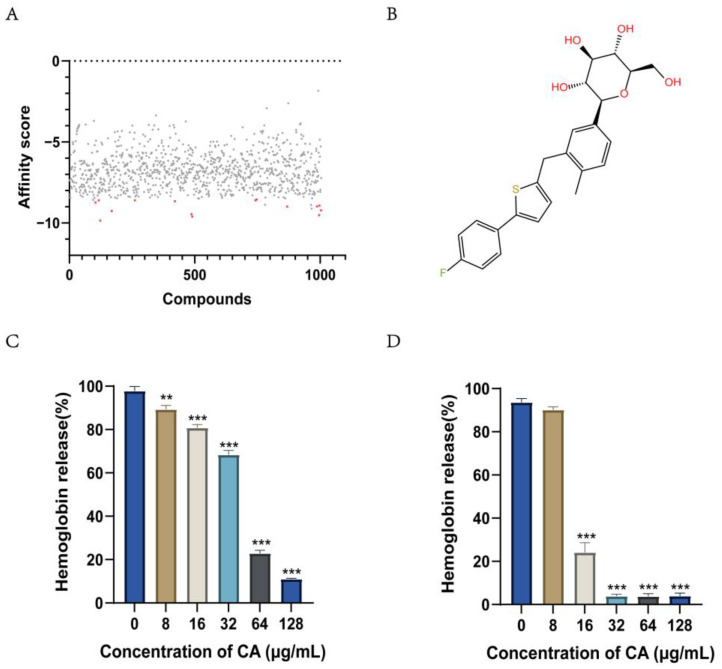
Canagliflozin inhibits hemolysin activity. (**A**) SLY was used as a target for virtual screening. A drug with a score of less than −0.85 was considered to be a hit and marked in red; (**B**) The chemical structure of canagliflozin; (**C**) Canagliflozin inhibited the hemolytic activity of the SC19 culture supernatant, ** *p* < 0.01, *** *p* < 0.001 vs. 0 μg/mL; (**D**) Canagliflozin inhibited the hemolytic activity of the SLY protein, *** *p* < 0.001 vs. 0 μg/mL. The data were obtained through three independent experiments. *p* values were determined by two-tailed unpaired *t* test.

**Figure 2 ijms-24-13074-f002:**
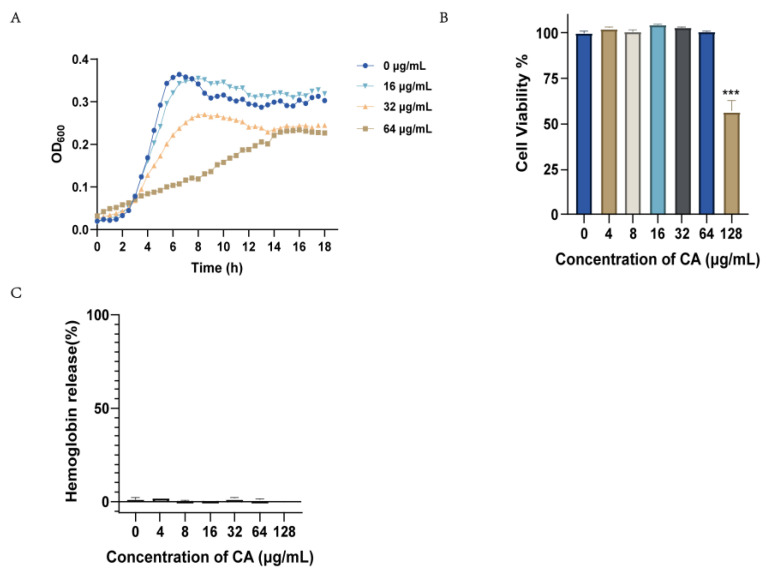
Assessment of the antimicrobial activity and safety of canagliflozin. (**A**) Growth curves of SC19 with different concentrations of canagliflozin (0, 16, 32, 64 μg/mL) were measured every half hour; (**B**) Cytotoxicity of canagliflozin for Vero cells *** *p* < 0.001 vs. 0 μg/mL; (**C**) Hemolytic toxicity of canagliflozin. All experiments were performed three times, and the mean ± SD is shown. *p* values were determined by two-tailed unpaired *t* test.

**Figure 3 ijms-24-13074-f003:**
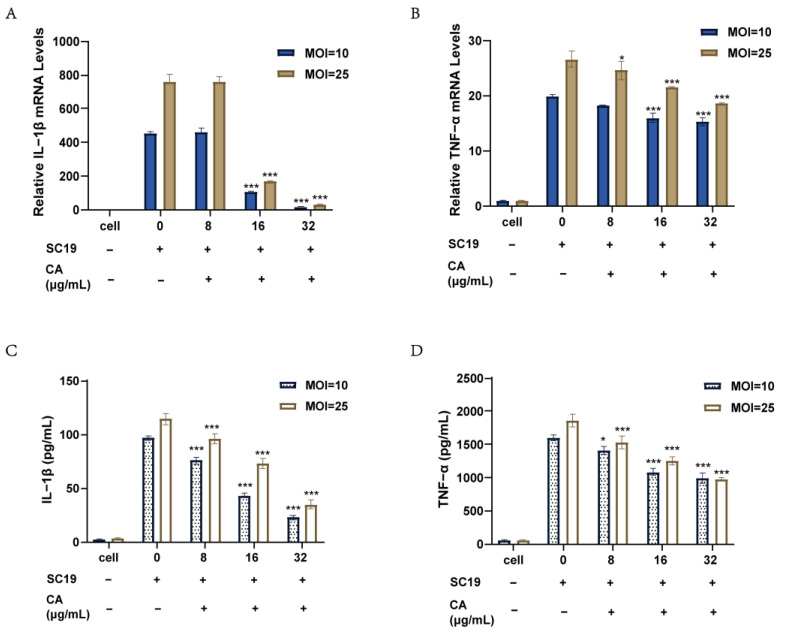
Canagliflozin reduced the levels of inflammatory factors induced by SC19 infection of RAW264.7 cell. RAW264.7 macrophages were infected with SC19 (MOI = 10 or 25) and treated with 0, 8, 16, or 32 μg/mL canagliflozin for 6 h. IL-Ιβ (**A**) and TNF-α (**B**) mRNA levels were determined by qRT-PCR. The protein levels of the cytokines IL-Ιβ (**C**) and TNF-α (**D**) in the supernatant were determined using a commercially available ELISA. All experiments were performed three times, and the mean ± SD is shown. *p* values were determined by two-way ANOVA, * *p* < 0.05, *** *p*  <  0.001 vs. 0 μg/mL.

**Figure 4 ijms-24-13074-f004:**
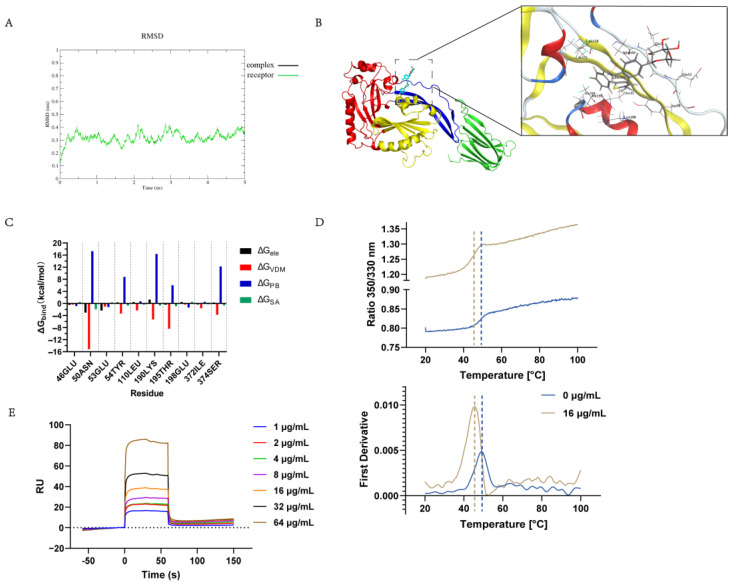
Identification of the binding of canagliflozin and hemolysin. (**A**) RMSD values of the SLY-canagliflozin complex and SLY during simulation; (**B**) 3D structure obtained by MD simulation. Domain 1 is marked in red, Domain 2 is marked in blue, Domain 3 is marked in yellow, and Domain 4 is marked in green; (**C**) Decomposition of the binding energies obtained for the binding sites of SLY-canagliflozin using the MM-PBSA method; (**D**) Determination of protein thermal stability of canagliflozin in SLY; (**E**) SPR binding curves for the interaction between SLY and canagliflozin. RU, resonance units (KD = 0.90 μM).

**Figure 5 ijms-24-13074-f005:**
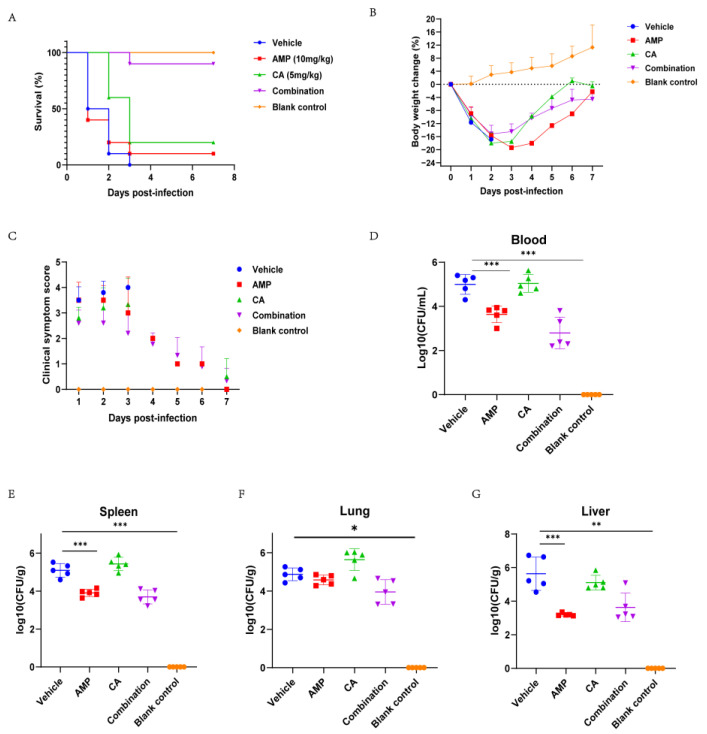
Canagliflozin increased the therapeutic effect of ampicillin on SC19 infection in mice. A dose of 2.5 × 10^8^ CFU was injected intraperitoneally into mice. (**A**) The survival rate of mice treated with canagliflozin, ampicillin, and the combination (*n* = 10 per group). (**B**) Effect of canagliflozin on body weight change in SC19-infected mice. The average change in body weight of the surviving mice was calculated daily. (**C**) Clinical signs of SC19-infected mice after treatment with canagliflozin, ampicillin, and the combination. Mice (*n* = 5) infected with a sublethal dose of SC19 (1 × 10^8^ CFU) for 6 h were treated with canagliflozin, ampicillin, and a combined group. The bacterial load in the blood (**D**), spleen (**E**), lung (**F**), and liver (**G**) of the mice was measured 12 h after infection. *p* values were determined by two-tailed unpaired *t* test; * *p* < 0.05, ** *p* < 0.01, *** *p* < 0.001 vs. vehicle.

**Figure 6 ijms-24-13074-f006:**
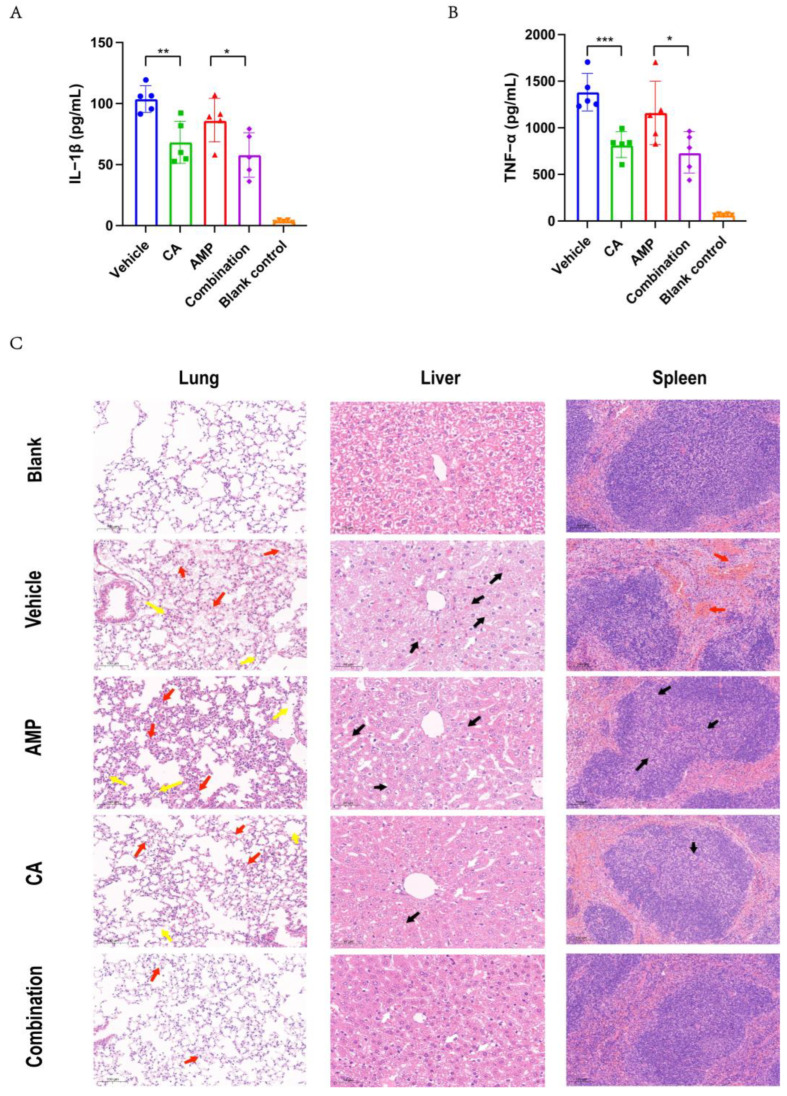
Inflammatory cytokines and histopathological damage in SC19-infected mice. Expression levels of IL-1β (**A**) and TNF-α (**B**) in infected mice. The group treated with canagliflozin alone showed a significant anti-inflammatory effect. Compared with ampicillin, the combination treatment significantly reduced the production of inflammatory cytokines. (**C**) Pathological changes in tissue sections of SC19-infected mice treated with canagliflozin for 6 h. Tissue congestion is indicated by red arrows, inflammatory cell infiltration by yellow arrows, necrosis by black arrows, and vacuolar degeneration by white arrows. *p* values were determined by two-tailed unpaired *t* test. (* *p* < 0.05, ** *p* < 0.01, *** *p* < 0.001; *n* = 5).

**Table 1 ijms-24-13074-t001:** The MIC value of canagliflozin against pathogenic bacteria.

Strain	MIC (μg/mL)	
	Canagliflozin	Ampicillin
*S. suis* SC19 ^2^	128	<1
*S. pneumoniae* ATCC 49619 ^1^	64	<1
*S. aureus* ATCC 25923 ^1^	64	<1
*S. aureus* USA300 ^1^	>128	64
*S. aureus* ATCC 43300 ^1^	64	8
*L. monocytogenes* ATCC19115 ^1^	64	2
*B. subtilis* NCD-2 ^1^	64	2
*S. suis* K6 ^2^	128	<1
*S. suis* 1422 ^2^	128	<1
*S. suis* 1519 ^2^	128	<1
*S. suis* 1808028 ^2^	128	<1
*S. suis* 1765 ^2^	128	64
*S. suis* 1801101 ^2^	>128	<1
*S. suis* 1804005 ^2^	>128	<1
*S. aureus* 1605020 ^2^	128	<1
*S. aureus* 2117 ^2^	128	64
*S. aureus* 1802097 ^2^	>128	<1

^1^ standard strains of pathogenic bacteria; ^2^ clinical strains of pathogenic bacteria.

## Data Availability

The data presented in this study are available on request from the corresponding author.

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
