# Peer review of "Canagliflozin Inhibited the Activity of Hemolysin and Reduced the Inflammatory Response Caused by Streptococcus suis"

_ijms, 2023, doi:10.3390/ijms241713074_

Round 1

Reviewer 1 Report

Article #Canagliflozin inhibited the activity of hemolysin and reduced 2 the inflammatory response caused by Streptococcus suis# is interesting but:

It is necessary to correct numerous grammatical and spelling errors in the text (e.g., italicise the names of microorganisms even 

in references; write the number separately from the unit);

number of references is 29 - too small;

please, separate Fig 2 A because it is unreadible

Reviewer 2 Report

The study by Li et al. describes the effects of canagliflozin, a hypoglycemic agent in countering the hemolytic and iflammatory effects of S. suis hemolysin and based on the data the potential repurposing of this drug in the treatment of Streptococcal toxic shock-like syndrome are discussed. The authors used a virtual screening approach from an FDA-approved drug library. The study is well-conducted, and the results are clearly described and interpreted. I have a few minor suggestions to improve the quality of the manuscript. Grammatical and typo errors are seen frequently in the text and should be carefully revised.

-        Did the authors measure monitor body temperature and changes in body weight of the mice? Were there any specific clinical scores assigned to determine the clinical symptoms and pre-survival status of the infected mice, with and without treatment? Please elaborate on this point in the methods and discuss it in the discussion.

-        Were any parameters used to estimate hemolysis in vivo? How would the authors link clinical efficacy in vitro with the in vivo observations?

-        Please expand on the discussion of the potential utilization of canagliflozin in human and livestock infections in future in the context of risks and benefits of the concentrations used in this study.

Please check for grammatical and spelling errors in the main text

Reviewer 3 Report

The authors attempted to find drugs to reduce the inflammatory effects of Streptococcus suis, using an in silico model followed by in vivo experiments.

Major issues

-The introduction is a bit lame and does not set the scene well. It must be extended and must be more convincing regarding the importance of the study.

-The use of only one strain of S. suis is a significant limitation of the study, given the wide differences that exist between strains of this organism. The authors must justify using only one strain.

-Please quantification scale for the pathology findings (e.g., presence and type of cells, presence of extravasation, tissue damage etc.) and then quantify please the effects of the drug on tissues.

Minor issues

-How did you prove normal distribution of results and performed parametric statistical tests?

Round 2

Reviewer 3 Report

The authors have made convincing and significant improvements in the revised manuscript, which is almost ready for publication, after only updating the list of references with some very recent relevant references (published just last month).

Author Response

Dear Reviewer:

We are very grateful to Reviewer for reviewing the paper so carefully. We have tried our best to improve and made some changes in the manuscript. The main corrections in the paper and the response to the reviewer’s comments are as flowing:

  • Updating the list of references with some very recent relevant references (published just last month).
  • Response: Thank you for this valuable feedback. We have revised and updated the references in the article.

We have included the most recent relevant references in the introduction to pathogens (page 1 starting line 35).

And with respect to the targeting of virulence proteins for drug screening, we add newly published references to the support of the argument (page 2 starting line 54).

We have updated to the description of metformin about the new progress in antimicrobial (page 2 starting line 62).

Thank you for your suggestions. All your suggestions are very important. They are of great guiding significance to my further thesis writing and scientific research.

Thank you and best regards.